# Do Language Models Plan Ahead for Future Tokens?

**Wilson Wu**
Department of Mathematics
University of Colorado Boulder
wiwu2390@colorado.edu

**John X. Morris**
Department of Computer Science
Cornell University
jxm3@cornell.edu

**Lionel Levine**
Department of Mathematics
Cornell University
levine@math.cornell.edu

## Abstract

Do transformers "think ahead" during inference at a given position? It is known transformers prepare information in the hidden states of the forward pass at time step $t$ that is then used in future forward passes $t + \tau$. We posit two explanations for this phenomenon: *pre-caching*, in which off-diagonal gradient terms present during training result in the model computing features at $t$ irrelevant to the present inference task but useful for the future, and *breadcrumbs*, in which features most relevant to time step $t$ are already the same as those that would most benefit inference at time $t + \tau$. We test these hypotheses by training language models without propagating gradients to past timesteps, a scheme we formalize as *myopic training*. In a constructed synthetic data setting, we find clear evidence for pre-caching. In the autoregressive language modeling setting, our experiments are more suggestive of the breadcrumbs hypothesis, though pre-caching increases with model scale.

## 1 Introduction

Humans are known to think ahead while speaking; decades of linguistics research (Huettig, 2015; Miller, 1951) have shown evidence that human language users internally predict upcoming language input, words and sometimes sentences ahead (Barthel et al., 2016).

Unlike humans, contemporary language models allocate a fixed amount of information processing for each token when "speaking" (Vaswani et al., 2017). Do language models, like humans, think ahead? Recent work (Pal et al., 2023; Hernandez et al., 2024) has shown that tokens beyond the immediate next token can be predicted by probing the hidden state of the language model. Model outputs at future tokens can be predicted to some extent using linear probes on model hidden states, and interventions on hidden states can predictably alter future outputs.[1]

These findings indicate that model activations at a given timestep are at least somewhat predictive of future outputs. However, it remains unclear why this might be: is this just a happenstance property of the data, or because the model is deliberately preparing information for future timesteps, at the expense of degrading performance on the current position?

We observe that gradients during training optimize weights for both the loss at the current token position as well as for tokens later in the sequence. We question to what extent current transformer weights dedicate resources to the current token vs. allocating it for future tokens.

---

[1]Code to reproduce our results can be found at https://github.com/wiwu2390/FutureGPT-public

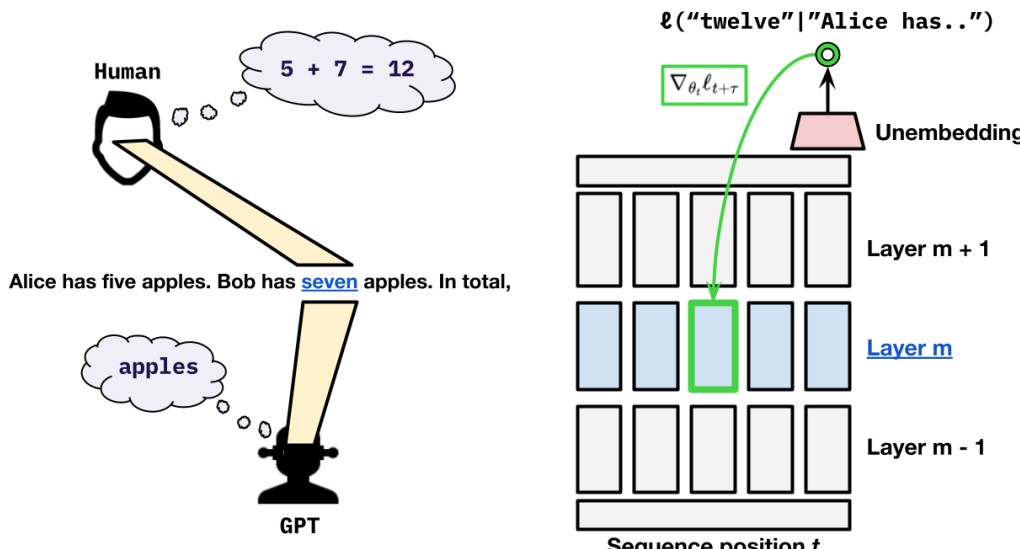

Figure 1: At which position is the computation required to correctly answer this math problem taking place? Cognitive science tells us that humans think ahead while speaking; we investigate the extent to which language models do the same.

We consider two possibilities: the *pre-caching* hypothesis, in which the transformer learns to compute features at time step $t$ that are irrelevant to the inference task at that current time step but may be useful for future time steps $t + \tau$, and the *breadcrumbs* hypothesis, in which the features most relevant to time step $t$ are already identical to those that would most benefit inference at time $t + \tau$. To evaluate which hypothesis might be correct, we propose a myopic training scheme that does not propagate gradients from the loss at the current position to hidden states from previous positions. We then evaluate the *myopia gap* in performance between myopically trained and vanilla transformers as a measure of pre-caching.

To consider whether language models might directly implement pre-caching, we design a synthetic scenario where the task can only be completed via explicit pre-caching. We configure a task where the model must precompute information for the next token, because otherwise the correct answer could not be accurately computed in a single forward pass. In this synthetic scenario, we find clear evidence that the transformer learns to pre-cache. When transformer-based sequence models must precompute information to minimize loss, they do so.

We then consider whether breadcrumbs or pre-caching is demonstrated in natural language models. Our experiments with myopic training suggest that, with small language models like GPT-2 (Radford et al., 2019), much less pre-caching occurs in this setting, pointing towards the breadcrumbs hypothesis. That is, we claim language models on this scale do not intentionally prepare information for the future to a significant extent. Instead, they compute features that are useful to predicting the immediate next token, which turn out to then be helpful at future steps; there is not a significant tradeoff between greedily optimizing for next token loss and ensuring future predictive performance.

However, we also find evidence that the importance of pre-caching increases with scale, becoming non-negligible with larger models, e.g. Pythia 2.8B (Biderman et al., 2023). This suggests that these larger models are "planning for the future" in a way that small models cannot.

## 2 Related work

**Future token meta-prediction.** Several recent works (nostalgebraist, 2020; Belrose et al., 2023; Pal et al., 2023; Cai et al., 2024) observe that transformer hidden states can be used to predict current and future tokens in a sequence, typically via linear probing. Notably, Hernandez et al. (2024) show that more complicated relationships are encoded linearly in hidden states, such as subject-object relations, implying that future tokens can also be predicted in specific cases. This future token predictivity has also been applied to speeding up inference by decoding future tokens in parallel (Stern et al., 2018; Cai et al., 2024). Unlike these works, we focus on the question of *how* the model learns to prepare hidden states that are useful for future prediction, possibly at the expense of current-token predictivity.

**Probing.** Our synthetic data experiments make use of *probing*, a technique where a simple auxiliary model is used to predict properties from target models' representations (Belinkov & Glass, 2019; Shi et al., 2016; Hewitt & Liang, 2019; Pimentel et al., 2020; Belinkov, 2021). Probing-based approaches are known to overestimate latent information if the classifier learns to do a task on its own (Belinkov, 2021), and probing analyses may only be informative when compared to probing a reasonable baseline (Hewitt & Liang, 2019). In our probing experiments, we avoid these pitfalls by ensuring that the function to be learned cannot possibly be computed by the probe itself.

**Mechanistic interpretability.** Our analysis of transformer models in a synthetic setting relates to the subfield of *mechanistic interpretability*, which seeks to understand models by isolating and explaining the behavior of their components (Olah et al., 2020; Bau et al., 2020; Meng et al., 2023; Nanda et al., 2023). Some of these works (Nanda et al., 2023; Li et al., 2023; Zhong et al., 2023) practice mechanistic interpretability by studying models trained on synthetic data. We apply some mechanistic interpretability techniques in a synthetic setting to study the problem of whether language models "think ahead" for future tokens. However, our approach also differs from that of mechanistic interpretability by analyzing the effect of the training procedure on the learned model.

## 3 Theory: Pre-caching or breadcrumbs?

Consider a generic causal sequence-to-sequence prediction task

$$(\mathbf{x}_1, \ldots, \mathbf{x}_n, \mathbf{y}_1, \ldots, \mathbf{y}_n) \sim \mathcal{D},$$

where $\mathcal{D}$ is a data distribution supported on $\mathbb{X}^n \times \mathbb{Y}^n$ for some domains $\mathbb{X}, \mathbb{Y}$. The task is to estimate the conditional expectations $\mathbb{E}_\mathcal{D}(\mathbf{y}_i \mid \mathbf{x}_1, \ldots, \mathbf{x}_i)$ for $1 \leq i \leq n$.[2] Note that we recover the autoregressive setting by setting $\mathbb{Y} = \mathbb{X}$ and $\mathbf{y}_i = \mathbf{x}_{i+1}$.

Transformer models trained on such tasks have been observed (Pal et al., 2023) to store information in hidden states during inference at position $i$ that is then used in future inference at $j > i$. However, since the loss associated with each step $i$ depends only on how well the model does at the immediate task of predicting $\mathbf{y}_i$, it may not be immediately obvious how this preparation for the future arises. We give names to two competing explanations:

- **Pre-caching:** The model "deliberately" computes and stores features that are expected to be useful for the future, even if they are irrelevant to the present.

- **Breadcrumbs:** The features that most benefit the present inference task are the same as those that are most useful to the future. When the model performs the present forward pass, it "unintentionally" leaves a trace ("breadcrumbs") that is then picked up by future passes.

---

[2]For classification tasks, we are typically interested in the conditional probabilities $\Pr_\mathcal{D}(\mathbf{y}_n = c \mid \mathbf{x}_1, \ldots, \mathbf{x}_n)$ for each class $c$. However, this can be subsumed into the generic case by letting $\mathbb{Y}$ be the probability simplex over all classes.

To disentangle these two explanations, we introduce a notion of *myopic* transformer models, which we show to be incapable of deliberate pre-caching—for these models, the extent to which past features are beneficial to the future is decided purely by the breadcrumbs explanation. Thus, the gap between vanilla and myopic transformer models is a quantitative measure of how much pre-caching is taking place.

### 3.1 Causal sequence modeling

Suppose, for the sake of exposition, that the transformer model $G$ uses independent parameters for each position. [3] Let $p$ be the parameter count of each forward pass of $G$. Then, letting $\boldsymbol{\theta}_i \in \Theta = \mathbb{R}^p$ be all parameters used by $G$ at position $i$, a transformer $G$ is a parameterized function

$$G \colon \mathbb{X}^n \times \Theta^n \to \mathbb{Y}^n, \qquad (\mathbf{x}_1, \ldots, \mathbf{x}_n; \boldsymbol{\theta}_1, \ldots, \boldsymbol{\theta}_n) \mapsto (\hat{\mathbf{y}}_1, \ldots, \hat{\mathbf{y}}_n).$$

For $1 \leq i \leq n$, let $G_i(\mathbf{x}_1, \ldots, \mathbf{x}_n) \in \mathbb{Y}$ be the output of $G$'s $i$th forward pass. Because of the causal masking within $G$, this depends only on $\mathbf{x}_1, \ldots, \mathbf{x}_i$ and $\boldsymbol{\theta}_1, \ldots, \boldsymbol{\theta}_i$. That is, with slight abuse of notation, we may write

$$\hat{\mathbf{y}}_i = G_i(\mathbf{x}_1, \ldots, \mathbf{x}_n; \boldsymbol{\theta}_1, \ldots, \boldsymbol{\theta}_n) = G_i(\mathbf{x}_1, \ldots, \mathbf{x}_i; \boldsymbol{\theta}_1, \ldots, \boldsymbol{\theta}_i).$$

### 3.2 Off-diagonal gradient terms

Now, letting $\mathcal{L} \colon \mathbb{Y} \times \mathbb{Y} \to \mathbb{R}_+$ be some choice of loss function, the expected loss $\ell$ of a transformer model with parameters $\boldsymbol{\theta}_1, \ldots, \boldsymbol{\theta}_n$ is

$$\ell(\boldsymbol{\theta}_1, \ldots, \boldsymbol{\theta}_n) := \mathbb{E}_{(\vec{\mathbf{x}}, \vec{\mathbf{y}}) \sim \mathcal{D}} \sum_{i=1}^n \mathcal{L}(G_i(\mathbf{x}_1, \ldots, \mathbf{x}_i; \boldsymbol{\theta}_1, \ldots, \boldsymbol{\theta}_i), \mathbf{y}_i) =: \sum_{i=1}^n \ell_i(\boldsymbol{\theta}_1, \ldots, \boldsymbol{\theta}_i),$$

the sum over $1 \leq i \leq n$ of the expected loss $\ell_i$ at position $i$. (We suppress the dependence on $G$ and $\mathcal{D}$ for concision.) In practice, we always tie the weights across position. That is, all $\boldsymbol{\theta}_i$ are set equal to the same $\boldsymbol{\theta} \in \Theta$. Then, by the chain rule,

$$\nabla_{\boldsymbol{\theta}} \ell(\boldsymbol{\theta}, \ldots, \boldsymbol{\theta}) = \sum_{i=1}^n \nabla_{\boldsymbol{\theta}_i} \ell(\boldsymbol{\theta}_1, \ldots, \boldsymbol{\theta}_n)\big|_{\boldsymbol{\theta}_1 = \ldots = \boldsymbol{\theta}_n = \boldsymbol{\theta}} = \sum_{i=1}^n \sum_{j=i}^n \nabla_{\boldsymbol{\theta}_i} \ell_j(\boldsymbol{\theta}_1, \ldots, \boldsymbol{\theta}_j)\big|_{\boldsymbol{\theta}_1 = \ldots = \boldsymbol{\theta}_n = \boldsymbol{\theta}},$$

a sum over an upper-triangular expected Jacobian "matrix". The off-diagonal terms $i < j$, corresponding to the expected gradient of the model's future loss at position $j$ with respect to its weights at position $i$, are the training signals that encourage pre-caching.

### 3.3 Measuring pre-caching: The myopia gap

We say a model is *myopic* when each forward pass $G_i$ optimizes only $\ell_i$ without regard for future $\ell_j$ at $j > i$. In the untied weights case, the right definition is then apparent.

**Definition 1.** *The parameters* $(\tilde{\boldsymbol{\theta}}_1, \ldots, \tilde{\boldsymbol{\theta}}_n) \in \Theta^n$ *are* **untied-myopic** *if they satisfy*

$$\tilde{\boldsymbol{\theta}}_i \in \arg\min_{\boldsymbol{\theta}_i} \ell_i(\tilde{\boldsymbol{\theta}}_1, \ldots, \tilde{\boldsymbol{\theta}}_{i-1}, \boldsymbol{\theta}_i) \qquad \forall i \in \{1, \ldots, n\}. \tag{1}$$

**Definition 2.** *Let* $\mathbb{M}$ *be the feasible set of the constraints in Equation 1. The* **untied myopia gap** *is the smallest possible gap between the expected loss attained by a myopic model and the optimal model:*

$$p^* := \min_{(\tilde{\boldsymbol{\theta}}_1, \ldots, \tilde{\boldsymbol{\theta}}_n) \in \mathbb{M}} \ell(\tilde{\boldsymbol{\theta}}_1, \ldots, \tilde{\boldsymbol{\theta}}_n) - \min_{(\boldsymbol{\theta}_1, \ldots, \boldsymbol{\theta}_n) \in \Theta^n} \ell(\boldsymbol{\theta}_1, \ldots, \boldsymbol{\theta}_n) \geq 0. \tag{2}$$

In the tied weights case, it is perhaps not immediately clear what the right definition of myopia should be. It does not suffice to simply constrain the minimizations in Equation 1 to $\tilde{\boldsymbol{\theta}}_1 = \ldots = \tilde{\boldsymbol{\theta}}_n$, since $\min_{\boldsymbol{\theta}} \ell_i(\boldsymbol{\theta}, \ldots, \boldsymbol{\theta})$ is optimizing for pre-caching (the dependence on arguments $j < i$) as well as the present inference (the dependence on argument $i$). Instead,

---

[3]For example, this is true of absolute position embedding weights.

the right notion is a choice of tied parameters such that the model is, aggregated over positions, optimal for the present task when conditioned on a fixed past. That is, forward passes do not compute features for the future if they can compute other features more beneficial to the present.

**Definition 3.** *The parameters $\tilde{\theta} \in \Theta$ are **(tied-)myopic** if they satisfy*

$$\tilde{\theta} \in \arg\min_{\theta \in \Theta} \sum_{i=1}^{n} \ell_i(\tilde{\theta}, \ldots, \tilde{\theta}, \theta). \tag{3}$$

*The **(tied) myopia gap** is then defined analogously to Definition 2.*

The *breadcrumbs hypothesis* states that the myopia gap is small—near-optimal performance can be attained even when each forward pass is computing features relevant to only its own immediate inference task, with no regard to pre-caching for the future.

If the breadcrumbs hypothesis does not hold, we say that the model is *pre-caching*. It is important to remember that the $\ell_i$ depend on a choice of transformer model $G$ and dataset $\mathcal{D}$. That is, breadcrumbs and pre-caching are properties of the model architecture and the data considered as a whole.

Although a small myopia gap reveals that one can do just as well without pre-caching, it does not say much about any specific model. To measure pre-caching within a given model, we examine the extent to which its parameters violate the myopia constraints.

**Definition 4.** *The **(tied) local myopia bonus** at $\theta^* \in \Theta$ is*

$$\hat{c}(\theta^*) := \max_{\theta \in \Theta} \sum_{i=1}^{n} \left( \ell_i(\theta^*, \ldots, \theta^*) - \ell_i(\theta^*, \ldots, \theta^*, \theta) \right) \geq 0.$$

For further interpretation of the myopia gap and myopia bonus, see Appendix A.

### 3.4 Myopic gradient descent

Our heuristic remark in Section 3.2, that the off-diagonal gradient terms are responsible for pre-caching, is justified by Theorem 13 below. It states that, given certain regularity conditions on the loss terms $\ell_i$, performing gradient descent with the off-diagonal terms removed results in a myopic model in the sense of Definition 3. We call this *myopic descent*.

For myopic descent to be stable in the tied-weights case, we need, roughly speaking, for the model to depend more on the parameters associated with the present forward pass than those from the past. This is a plausible condition—dependence on the past is mediated purely by the attention mechanism, while the present forward pass depends on both attention and feedforward parameters. The precise condition we use is **forward bias** (Definition 11); see Appendix C for details and the proof of the theorem.

**Theorem 13.** *Let $f(\tilde{\theta}, \theta) := \sum_{i=1}^{n} \ell_i(\tilde{\theta}, \ldots, \tilde{\theta}, \theta)$. If $f$ is forward-biased, $\sigma$-strongly convex, and $L$-smooth, then, for some step size $\eta > 0$, the iterates of myopic descent with tied weights*

$$\theta^{(t+1)} = \theta^{(t)} - \eta \nabla_\theta f(\tilde{\theta}, \theta)\big|_{\tilde{\theta} = \theta = \theta^{(t)}}$$

*converge to $\tilde{\theta} \in \Theta$ satisfying the myopia constraints of Equation 3.*

## 4 Synthetic data experiments

### 4.1 The $\mathcal{D}_p$ task

To demonstrate a simple example where the transformer model learns to pre-cache (and thus the myopia gap is large), we construct the following synthetic dataset.

**Definition 5.** *The data distribution $\mathcal{D}_{p,a,b}^{N}$ is defined as the joint distribution of real-valued random variables $(x_n)_{n=1}^{N}, (y_n)_{n=1}^{N}, (z_n)_{n=1}^{N}$ where, for each $n$,*

- $x_n \sim \mathcal{N}(0,1)$ *(standard Gaussian)*

- $z_n \sim \text{Ber}(p)$ *(Bernoulli with probability $p$)*

- $y_n = z_n \sum_{i=1}^{a} \sin(bx_{n-i}) + (1 - z_n)x_n$

*and $\{x_n\}_{n\in\mathbb{N}} \cup \{z_n\}_{n\in\mathbb{N}}$ are mutually independent. In our experiments, we always set the parameters $a = b = 10$ and $N = 64$, so for convenience notate $\mathcal{D}_p := \mathcal{D}_{p,10,10}^{64}$.*

The intuition is that a transformer regression model $G$ trained on $\mathcal{D}_p$ would benefit from pre-caching $\sin(bx_n)$ during its forward pass at position $n$, even though this computation is irrelevant to its task of predicting $y_n$. One simple strategy that makes use of this pre-caching is Algorithm 1. [4]

The motivation for the Bernoulli variables $z_i$ is that, as $p$ decreases, the expected first time when $\sin(bx_n)$ becomes useful advances further into the future. In addition, when $p$ is sufficiently small, the probability $(1 - p)^a$ that the value $\sin(bx_n)$ is never useful at all becomes non-negligible. We will show that, even in this case, the transformer model learns to pre-cache.

**Investigating myopia.** Suppose that we train a myopic model (Section 3.4) on the same task. Since this model lacks off-diagonal gradient terms, we do not expect it to learn to pre-cache $\sin(bx_n)$ at position $n$. One possible strategy that does not use pre-caching is Algorithm 2. We expect this brute force algorithm to perform significantly worse given the same parameter count—it computes an $a$-dimensional nonlinear function within a single layer, while each layer of Algorithm 1 computes only scalar nonlinear functions.[5]

---

**Algorithm 1** Pre-caching algorithm

At position $n$,
    **input** $x_n, z_n$
    **layer 1 compute** $F_n := \sin(bx_n)$
    **layer 2 read** $F_{n-i}$ for $i = 1, \ldots, a$.
    **layer 2 compute**
    $\hat{y}_n := z_n \sum_{i=1}^{a} F_{n-i} + (1 - z_n)x_n$.
    **return** $\hat{y}_n$.

---

**Algorithm 2** Brute force algorithm

At position $n$,
    **input** $x_n, z_n$
    **layer 1 compute** $\varnothing$
    **layer 2 read** $x_{n-i}$ for $i = 1, \ldots, a$.
    **layer 2 compute**
    $\hat{y}_n := z_n \sum_{i=1}^{a} \sin(bx_{n-i}) + (1 - z_n)x_n$.
    **return** $\hat{y}_n$.

---

### 4.1.1 Evaluation: linear probing

To determine if the transformer model is computing $\sin(bx_n)$ at position $n$, we fit linear probes on the hidden states. We additionally compute the correlations between $\sin(bx_n)$ and each individual dimension (i.e., each neuron) of each hidden state.[6] See Section D.1.1 for details.

---

[4]We think of each transformer layer as a **read** operation, performed by the attention mechanism, followed by a **compute** operation, performed by the feedforward block.

[5]For example, Shen et al. (2022) provide upper bounds on MLP error that degrade exponentially in dimensionality given a fixed parameter and layer count.

[6]Note that, in order for linear probing to be meaningful, we must first ensure that there is no pre-existing linear relationship between the inputs and the quantities we are probing for. Since the $(x_n, z_n)_n$ are mutually independent, this follows from Lemma 15, stating that $x_n$ and $\sin(bx_n)$ have near-zero correlation for large enough $b$. In our experiments, we set $b = 10$, in which case $\rho(x_n, \sin(bx_n)) < 10^{-20}$. In other words, there is low predictive $\mathcal{V}$-information from the inputs to the target $\sin(bx_n)$, where $\mathcal{V}$ is the class of linear models (Xu et al., 2020).

### 4.1.2 Results for $\mathcal{D}_p$

For varying $p$, we train two-layer transformer models with embedding dimensions of 128 on $\mathcal{D}_p$ using using both ordinary and myopic gradient descent. Full architecture and training details are provided in Section D.1.

Examining the performance of each linear probe against $\sin(bx_{n-i})$ for varying $i$, we find strong evidence that the transformer model with vanilla training is indeed pre-caching $\sin(bx_n)$, possibly in order to implement Algorithm 1. Indeed, in Figure 2,

- The zeroth hidden state (i.e., the sum of the input and position embeddings) at position $n$ is correlated with only $x_n$.

- The first hidden state is correlated with $\sin(bx_n)$ but not correlated with any $\sin(bx_{n-i})$ for $i > 0$.

- The second hidden state (immediately before the output unembedding) is correlated with $\sin(bx_{n-i})$ for each $0 \leq i \leq a$.

Further, looking at the per-neuron correlations in Figure 3, we see that $\sin(bx_{n-i})$ for $1 \leq i \leq a$ are all correlated with a single 1-d subspace of the second hidden state (they share the same striping pattern); this is the subspace storing $\sum_{i=1}^{a} \sin(bx_{n-i})$. Meanwhile, $\sin(bx_n)$, as well as many of the $x_{n-i}$, are located in various other 1-d subspace of the second hidden state; these terms are all left over in the residual stream from previous layers, and are cleaned up only by the output unembedding.

On the other hand, in Table 1, the myopic models perform significantly worse. The per-neuron correlations in Figure 3 suggest that the myopic model may be implementing a

| $p$ | **Vanilla** | **Myopic** |
|------|---------|--------|
| 0.01 | 0.096   | 1.10   |
| 0.1  | 0.016   | 0.97   |
| 0.3  | 0.0030  | 1.03   |
| 1.0  | 0.0074  | 1.26   |

Table 1: Normalized Huber loss $\mathcal{L}/p$ for vanilla and myopic models trained and evaluated on $\mathcal{D}_p$ for each $p$ in our synthetic setting. For reference, the trivial model that always outputs zero attains a Huber loss of **1.26**.

crude approximation of Algorithm 2. This suggests that the synthetic setting has an inherently high *myopia gap*—it is impossible for the transformer model to do well without pre-caching.

### 4.2 Multiplication

In addition to the above $\mathcal{D}_p$ synthetic data setting, we also measure the myopia gap on the task of natural number multiplication. In particular, we find evidence suggesting that pre-caching is responsible for model computation on filler tokens, in the sense of Pfau et al. (2024). See Appendix D.3 for details.

## 5 Natural language experiments

### 5.1 GPT-2's myopia gap

In order to measure the extent to which transformer models learn to pre-cache on natural language data, we estimate both the myopia gap (Definition 3) in this setting as well as the local myopia bonus (Definition 4) of a transformer model with vanilla pre-training. Experiments in this subsection use the 124M-parameter GPT-2 architecture; see Table 4 in Appendix D for configuration details.

We train all models (vanilla and myopic) from random initialization for one epoch on 4.6M sequences from the MS MARCO dataset (Nguyen et al., 2016), truncated to length 64. To estimate the local myopia bonus of the vanilla model, we train another model from random initialization with the same architecture, but with past hidden states sourced from the frozen

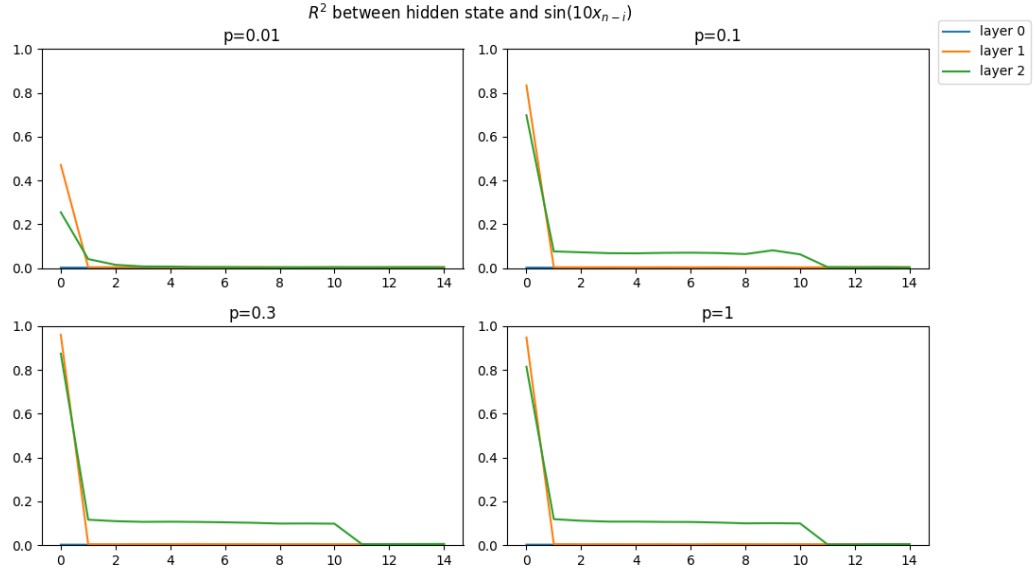

Figure 2: Empirical $R^2$ between linear probes fit on each layer of vanilla transformer models trained on $\mathcal{D}_p$ for $p \in \{0.01, 0.1, 0.3, 1\}$ to targets $\sin(b\mathbf{x}_{n-i})$. Computed over $50\,000$ samples from $\mathcal{D}_1$.

vanilla model during both training and evaluation.[7] See Appendix B for implementation details.

As baseline, we also train a "transformer bigram" model, a model with an identical architecture but all off-diagonal key/value states zeroed out.

### 5.1.1 GPT-2 results

From Table 2, the estimated myopia gap in this setting is $3.40 - 3.28 = \mathbf{0.12}$ cross entropy, while the local myopia bonus of the vanilla model is $3.28 - 3.26 = \mathbf{0.02}$.

The nonzero myopia gap suggests that pre-caching may provide a small positive benefit. Indeed, in Figure 4, we see that the myopic model outperforms the vanilla model at the beginning of the sequence, since it can allocate all compute to next-token prediction, but quickly falls behind as the length of the past increases, since it suffers from a lack of pre-cached information from earlier forward passes.[8]

However, note that this gap is much smaller than that between the vanilla model and the transformer bigram model (Table 2). That is, the myopic model is still able to leverage past information (breadcrumbs) to a significant extent, even if they optimized only for the present inference task. That the local myopia gap is near zero further supports this direction—the model learned through vanilla training does not trade off significantly between features useful for the present and pre-caching for the future.

| Model | Cross-entropy |
|---|---|
| Vanilla | 3.28 |
| Myopic | 3.40 |
| Local myopic | 3.26 |
| Transformer bigram | 5.33 |

Table 2: Validation cross-entropy loss obtained by GPT-2 with vanilla and myopic training

---

[7]Note that this "local myopic" model attains slightly *better* performance than the vanilla model; each forward pass can focus purely on next-token prediction, since past hidden states are supplied by a separate model.

[8]We use a sliding window over PG-19 (Rae et al., 2019) samples, which comprise longer sequences, in order to reduce noise in our per-position loss estimates. The distribution shift between MS MARCO and PG-19 does not affect the relative comparison of myopic and vanilla GPT-2.

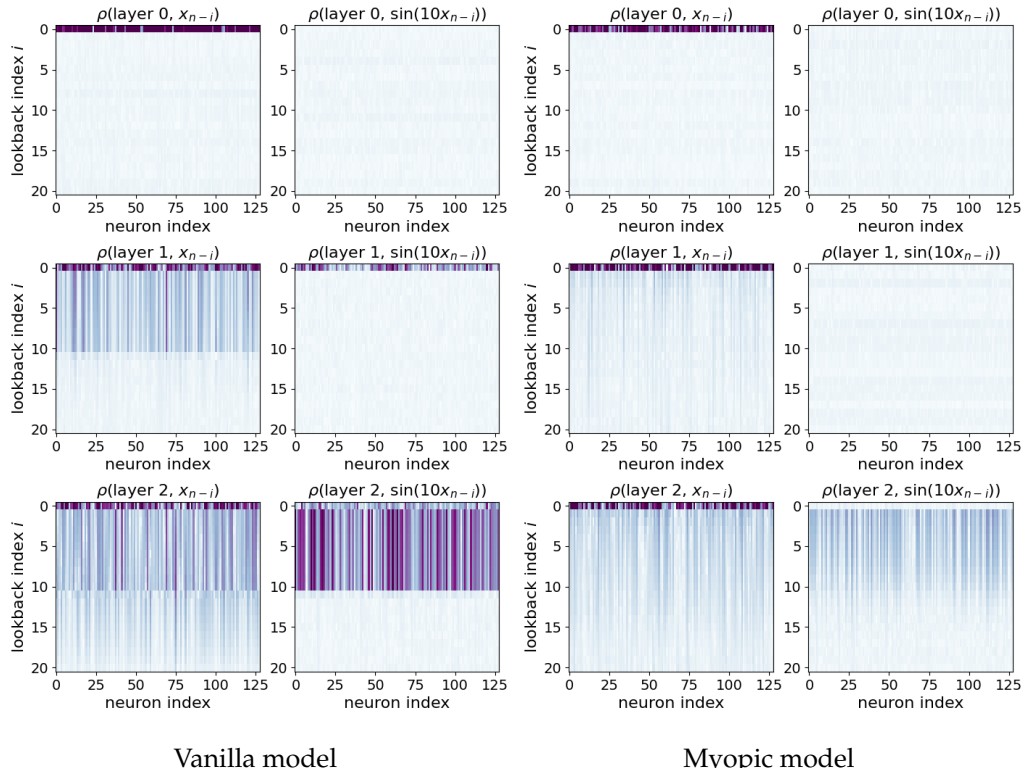

Vanilla model               Myopic model

Figure 3: Empirical correlations between each hidden state neuron and $x_{n-i}$ or $\sin(bx_{n-i})$. Models are vanilla (left two columns) and myopic (right two columns) transformers trained on $\mathcal{D}_{0.3}$.

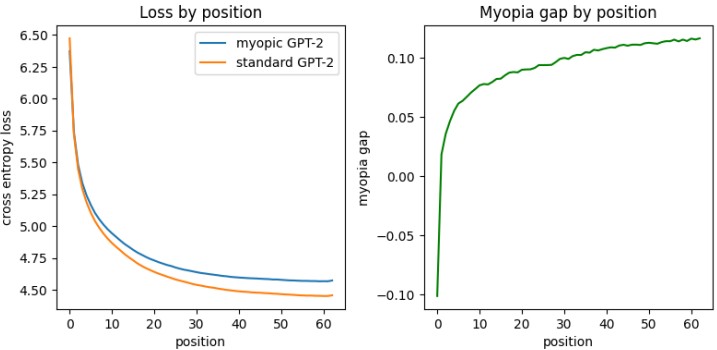

Figure 4: Cross-entropy loss of vanilla and myopic GPT-2 models by token position, and their difference. Evaluated on a sliding window over a 100K-token sample text from the PG-19 dataset (Rae et al., 2019). Aggregate cross-entropy losses on this sample are 4.67 (vanilla) and 4.77 (myopic).

## 5.2 Myopia gap scaling

One might suppose that the relatively small myopia gap of GPT-2 is due to the relative simplicity of the small architecture we consider, and that larger language models might exhibit a more pronounced myopia gap.

To test this, we train both vanilla and myopic transformers from the Pythia LLM suite (Biderman et al., 2023), ranging in size from 14M to 2.8B parameters, on one epoch of 10M

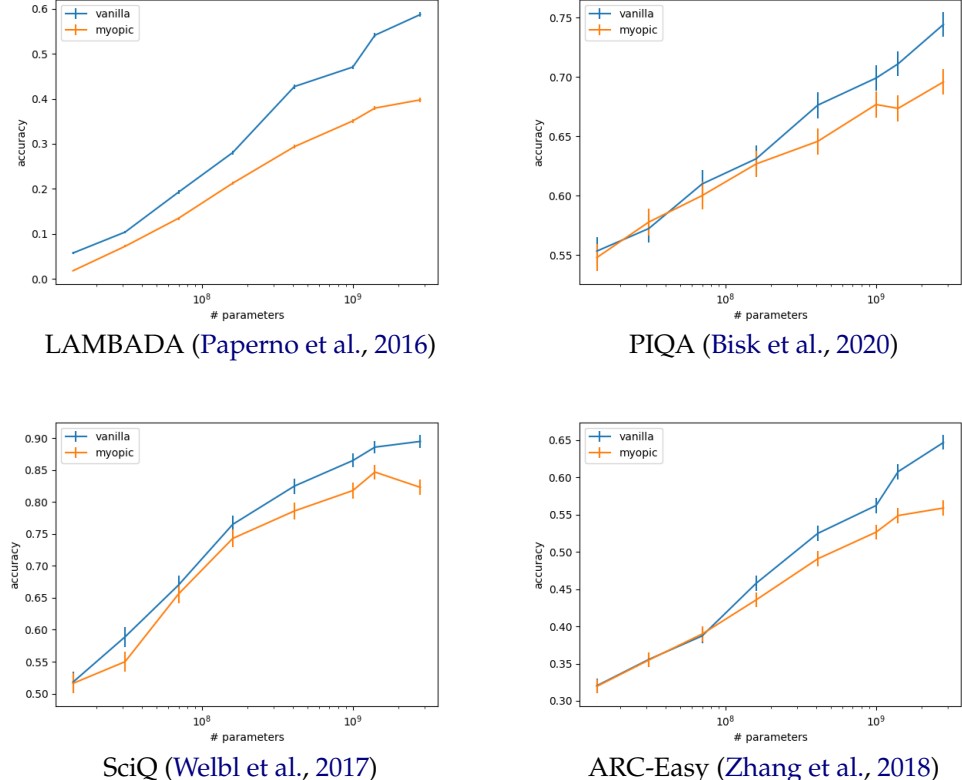

Figure 5: Benchmarks of Pythia models fine-tuned on the Pile dateset using vanilla and myopic descent.

sequences of 64 tokens each subsampled from the Pile dataset (Gao et al., 2020). (We use the same subsampled dataset for every training run.) We report validation cross-entropy loss (Figure 7 in Appendix D) as well as performance on a variety of natural language experiments (Figure 5). Note that, unlike in the GPT-2 experiments (Section 5.1), which start from random initialization, we start all training for Pythia models from the pre-trained checkpoints provided by Biderman et al. (2023)—for the larger architectures, the 10M sequence dataset we use is not sufficiently large to use for pre-training from random initialization.

## 6 Discussion and future work

Using a synthetic dataset, we demonstrate that pre-caching can indeed be learned by a transformer model. On the other hand, our experiments with natural language suggest that the breadcrumbs hypothesis is more explanatory for that setting, especially with smaller models, but that the importance of pre-caching increases with scale.

If the myopia gap is indeed not large in practice, there may be several applications of myopic training. We hypothesize that myopic transformers may have advantages in terms of safety and/or interpretability—it may be easier to understand what a model is doing if we know that everything that it is computing on each forward pass is directly towards the goal of predicting the immediate next token. For example, as seen in Appendix D.3, myopic models may not be able to make use of computation on the forward passes of filler tokens in the sense of Pfau et al. (2024).

Another possibility is that of automatically swapping in a locally myopic model (Section 5.1) on forward passes where we detect it is beneficial to sacrifice future performance in favor of immediate next-token accuracy (for example, on especially important tokens, or near the end of a text). We leave these possible applications to future work.

## Acknowledgments

LL thanks Lukas Berglund and David Schneider-Joseph for inspiring conversations. This research was partly supported by Open Philanthropy and the Berkeley Existential Risk Initiative.

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

## A  Myopia bonus and malus

Notice that the myopia gap consists of two pieces: a *myopia bonus*, the improvement that can be obtained at the current forward pass by ignoring the future forward passes; and a *myopia malus*, the cost to the future forward passes that is incurred by not pre-caching for them. To be precise, in the untied case,[9] given a choice of myopic $\tilde{\boldsymbol{\theta}}_1, \ldots, \tilde{\boldsymbol{\theta}}_n$ satisfying constraints (1) of Definition 1, write

$$\ell(\tilde{\boldsymbol{\theta}}_1, \ldots, \tilde{\boldsymbol{\theta}}_n) - \min_{\boldsymbol{\theta}_1, \ldots, \boldsymbol{\theta}_n} \ell(\boldsymbol{\theta}_1, \ldots, \boldsymbol{\theta}_n)$$

$$= \sum_{i=1}^{n} \left( \min_{\boldsymbol{\theta}_{i+1}, \ldots, \boldsymbol{\theta}_n} \ell(\tilde{\boldsymbol{\theta}}_1, \ldots, \tilde{\boldsymbol{\theta}}_{i-1}, \tilde{\boldsymbol{\theta}}_i, \boldsymbol{\theta}_{i+1}, \ldots, \boldsymbol{\theta}_n) - \min_{\boldsymbol{\theta}_i, \ldots, \boldsymbol{\theta}_n} \ell(\tilde{\boldsymbol{\theta}}_1, \ldots, \tilde{\boldsymbol{\theta}}_{i-1}, \boldsymbol{\theta}_i, \boldsymbol{\theta}_{i+1}, \ldots, \boldsymbol{\theta}_n) \right)$$

$$= \sum_{i=1}^{n} \left( \ell_i(\tilde{\boldsymbol{\theta}}_1, \ldots, \tilde{\boldsymbol{\theta}}_{n-1}, \tilde{\boldsymbol{\theta}}_i) - \ell_i(\tilde{\boldsymbol{\theta}}_1, \ldots, \tilde{\boldsymbol{\theta}}_{n-1}, \boldsymbol{\theta}_i^{*i}) \right)$$

$$+ \sum_{i=1}^{n} \sum_{j=i+1}^{n} \left( \ell_j(\tilde{\boldsymbol{\theta}}_1, \ldots, \tilde{\boldsymbol{\theta}}_{i-1}, \tilde{\boldsymbol{\theta}}_i, \boldsymbol{\theta}_{i+1}^{*i+1} \ldots, \boldsymbol{\theta}_n^{*i+1}) - \ell_j(\tilde{\boldsymbol{\theta}}_1, \ldots, \tilde{\boldsymbol{\theta}}_{i-1}, \boldsymbol{\theta}_i^{*i}, \boldsymbol{\theta}_{i+1}^{*i}, \ldots, \boldsymbol{\theta}_n^{*i}) \right).$$

where we define

$$\boldsymbol{\theta}_i^{*i}, \ldots, \boldsymbol{\theta}_n^{*i} \in \operatorname*{arg\,min}_{\boldsymbol{\theta}_i, \ldots, \boldsymbol{\theta}_n} \ell(\tilde{\boldsymbol{\theta}}_1, \ldots, \tilde{\boldsymbol{\theta}}_{i-1}, \boldsymbol{\theta}_i, \boldsymbol{\theta}_{i+1} \ldots, \boldsymbol{\theta}_n).$$

That is, the myopia gap is the sum $c + d = \sum_i c_i + \sum_i d_i$ of the myopia bonuses

$$c_i(\tilde{\boldsymbol{\theta}}_1, \ldots, \tilde{\boldsymbol{\theta}}_n) := \ell_i(\tilde{\boldsymbol{\theta}}_1, \ldots, \tilde{\boldsymbol{\theta}}_{n-1}, \tilde{\boldsymbol{\theta}}_i) - \ell_i(\tilde{\boldsymbol{\theta}}_1, \ldots, \tilde{\boldsymbol{\theta}}_{n-1}, \boldsymbol{\theta}_i^{*i}) \leq 0,$$

---

[9]The tied case is analogous, so we do not write it explicitly.

(the inequality following from the myopia constraints (1)), and the myopia maluses

$$d_i(\tilde{\boldsymbol{\theta}}_1, \ldots, \tilde{\boldsymbol{\theta}}_n)$$
$$:= \sum_{j=i+1}^{n} \left( \ell_j(\tilde{\boldsymbol{\theta}}_1, \ldots, \tilde{\boldsymbol{\theta}}_{i-1}, \tilde{\boldsymbol{\theta}}_i, \boldsymbol{\theta}_{i+1}^{*i+1} \ldots, \boldsymbol{\theta}_n^{*i+1}) - \ell_j(\tilde{\boldsymbol{\theta}}_1, \ldots, \tilde{\boldsymbol{\theta}}_{i-1}, \boldsymbol{\theta}_i^{*i}, \boldsymbol{\theta}_{i+1}^{*i}, \ldots, \boldsymbol{\theta}_n^{*i}) \right)$$
$$\geq 0,$$

with $d_i + c_i \geq 0$ for each $1 \leq i \leq n$ by the definition of the $\boldsymbol{\theta}_j^{*i}$. *A priori*, a small myopia gap does not necessarily imply a small (in magnitude) myopia bonus $c$ and malus $d$. Indeed, in the case when the myopia gap is small, a large value for $c$ (and thus a corresponding large value for $d$) means precisely that the transformer model is committing significant resources to pre-caching that could otherwise have been used to improve inference on the current position. On the other hand, it is possible that both $c$ and $d$ are small; that is, there is not much cost associated with pre-caching for the future, as the present forward pass already results in information (breadcrumbs) useful for that purpose.

In practice, the myopia bonus may be difficult to estimate, as it depends on the result of $O(n)$ separate optimization problems (each of which, in practice, is a full transformer model training run). Thus, we instead compute the local myopia bonus of Definition 4.

### A.1 Explicit gradient paths

The dependence of forward pass $G_i$ on previous forward passes $G_j$ for $j < i$ is mediated through hidden states $\boldsymbol{h}_1, \ldots, \boldsymbol{h}_{i-1}$:

$$G_i(\mathbf{x}_1, \ldots, \mathbf{x}_i; \boldsymbol{\theta}_1, \ldots, \boldsymbol{\theta}_i) = \tilde{G}_i(\boldsymbol{h}_1, \ldots, \boldsymbol{h}_{i-1}, \mathbf{x}_i; \boldsymbol{\theta}_i)$$

where the $\boldsymbol{h}_i$ are themselves recursively defined parameterized functions $\boldsymbol{h}_i = \boldsymbol{h}_i(\boldsymbol{h}_1, \ldots, \boldsymbol{h}_{i-1}, \mathbf{x}_i; \boldsymbol{\theta}_i)$. (Note that we are making a choice here to consider transformers as functions of hidden states, and not their key/value states. This has implications for how myopic descent is defined: when hidden state $\boldsymbol{h}_j$ is attended to by forward pass $i > j$, we consider the key and value weights $W_K$ and $W_V$, respectively, to belong to forward pass $i$. Thus, they are updated by the gradient wrt. $\ell_i$.)

With the hidden states explicitly written out, the gradient wrt the loss is a sum over all possible paths to the present: for $j < i$,

$$\frac{\partial G_i}{\partial \boldsymbol{\theta}_j}(\mathbf{x}_1, \ldots, \mathbf{x}_i; \boldsymbol{\theta}_1, \ldots, \boldsymbol{\theta}_i) = \sum_{j=i_1 < \ldots < i_m < i} \frac{\partial \boldsymbol{h}_j}{\partial \boldsymbol{\theta}_j} \frac{\partial \hat{G}_i}{\boldsymbol{h}_{i_m}} \prod_{k=1}^{m-1} \frac{\partial \boldsymbol{h}_{i_k}}{\partial \boldsymbol{h}_{i_{k-1}}}.$$

where the sum is over all partitions $j = i_1 < \ldots < i_m < i$.

## B  The myopic attention mechanism

An important primitive that we use when implementing the myopic gradient descent of Section 3.4, the local myopic bonus of Definition 4, and the transformer bigram in Section 5.1 is an attention mechanism that uses key/value states for past forward passes differing from those it uses for the current pass, while still computing all forward passes in parallel. We call our construction the *myopic attention mechanism*. We use it to implement several distinct transformer training methodologies:

- When training with *myopic descent*, the past key/value states are the result of key/value weights $W_K$ and $W_V$, respectively, multiplying a cloned and detached copy of the previous hidden states.

- During both training and inference of a *local myopic model*, the past key/value states come from a separate frozen pre-trained transformer model.

- During both training and inference of a *transformer bigram model*, the past key/value states are simply zeroed out.

Let $X = (x_1, \ldots, x_n)^\top \in \mathbb{R}^{n \times d}$ be the sequence of residual stream hidden states per position, with each row representing one position's hidden state in $\mathbb{R}^d$. Let $W_Q, W_K, W_V \in \mathbb{R}^{d \times h}$ be the query, key, value weight matrices for one attention head, of dimensionality $h$, in the transformer $G$. Denote

$$Q := XW_Q, \; K := XW_K, \; V := XW_V$$

and let $\tilde{Q}, \tilde{K}, \tilde{V}$ be the alternate states we wish to use for off-diagonal attention terms. We adopt the convention that lowercase letters with subscripts represent rows of matrices; e.g. $q_i$ is the $i$th row of $Q$. For simplicity of presentation, we omit causal masking; the modifications that should be made in the presence of a mask are straightforward.

Recall that the vanilla attention mechanism for $G$ is

$$Y = \sigma(QK^\top)V,$$

where $\sigma$ is row-wise softmax. Writing this out token-wise,

$$y_i = Z_i^{-1} \sum_{j=1}^n \exp(q_i^\top k_j) \mathbf{v}_j,$$

where $Z_i$ is the partition function

$$Z_i := \sum_{j=1}^n \exp(q_i^\top k_j).$$

The myopic attention mechanism, on the other hand, is written tokenwise as

$$
\begin{aligned}
\tilde{y}_i &= \tilde{Z}_i^{-1} \left( \exp(q_i^\top k_i) \mathbf{v}_i + \sum_{j \neq i} \exp(q_i^\top \tilde{k}_j) \tilde{\mathbf{v}}_j \right) \\
&= \tilde{Z}_i^{-1} \sum_{j=1}^n \exp(q_i^\top \tilde{k}_j + \delta_{ij} q_i^\top (k_j - \tilde{k}_j))(\tilde{\mathbf{v}}_j + \delta_{ij}(\mathbf{v}_j - \tilde{\mathbf{v}}_j)) \\
&= \sum_{j=1}^n a_{ij} \tilde{\mathbf{v}}_j - \sum_{j=1}^n \delta_{ij} a_{ij}(\mathbf{v}_j - \tilde{\mathbf{v}}_j)
\end{aligned}
$$

where

$$
\begin{aligned}
\tilde{Z}_i &:= \exp(q_i^\top k_i) + \sum_{j \neq i} \exp(q_i^\top \tilde{k}_j) \\
&= \sum_{j=1}^n \exp(q_i^\top \tilde{k}_j + \delta_{ij} q_i^\top (k_j - \tilde{k}_j)), \\
a_{ij} &:= \tilde{Z}_i^{-1} \exp(q_i^\top \tilde{k}_j + \delta_{ij} q_i^\top (k_j - \tilde{k}_j)),
\end{aligned}
$$

and $\delta_{ij}$ is the Kronecker delta. Now, notate

$$(\operatorname{diag} A)_{ij} := \delta_{ij} A_{ij}.$$

That is, $\operatorname{diag} A$ is the diagonal matrix that has the same entries as $A$ along the diagonal and is zero elsewhere. We are now able to write the myopic attention mechanism in matrix form:

$$\tilde{Y} = A\tilde{V} + (\operatorname{diag} A)(V - \tilde{V}),$$

where

$$A := \sigma(Q\tilde{K}^\top + \operatorname{diag}(Q(K^\top - \tilde{K}^\top))).$$

## C   Proofs

We use two assumptions on the loss function to be minimized. These are standard in the first-order methods literature.

**Definition 6.** *A function $f: \mathbb{R}^n \to \mathbb{R}$ is called L-**smooth** if it is continuously differentiable with L-Lipschitz gradient. That is, for all $\boldsymbol{x}, \boldsymbol{y}$ in the domain,*

$$\|\nabla f(\boldsymbol{x}) - \nabla f(\boldsymbol{y})\|_2 \leq L\|\boldsymbol{x} - \boldsymbol{y}\|_2.$$

**Definition 7.** *A function $f: \mathbb{R}^n \to \mathbb{R}$ is called $\sigma$-**strongly convex** if, for all $\boldsymbol{x}, \boldsymbol{y}$ in the domain,*

$$f(\boldsymbol{y}) \geq f(\boldsymbol{x}) + \nabla f(\boldsymbol{x})^\top (\boldsymbol{y} - \boldsymbol{x}) + \frac{\sigma}{2}\|\boldsymbol{y} - \boldsymbol{x}\|_2^2.$$

In particular, recall that strong convexity implies the existence of a unique minimum. Hence, it makes sense to write, for example, $\boldsymbol{x}^* = \arg\max_{\boldsymbol{x}} f(\boldsymbol{x})$ without ambiguity.

### C.1 Gradient descent with untied weights

**Theorem 8.** *Assume $\ell: \Theta^n \to \mathbb{R}$ is $\sigma$-strongly convex and L-smooth for some $\sigma, L > 0$. Consider ordinary gradient descent with untied weights*

$$\boldsymbol{\theta}_i^{(t)} = \boldsymbol{\theta}_i^{(t-1)} - \eta \nabla_{\boldsymbol{\theta}_i} \ell(\boldsymbol{\theta}_1^{(t-1)}, \ldots, \boldsymbol{\theta}_i^{(t-1)}) \qquad \forall i \in \{1, \ldots, n\}.$$

*Then, for $\boldsymbol{\theta}_1^*, \ldots, \boldsymbol{\theta}_n^* = \arg\min_{\boldsymbol{\theta}_1, \ldots, \boldsymbol{\theta}_n} \ell(\boldsymbol{\theta}_1, \ldots, \boldsymbol{\theta}_n)$, for small enough $\eta > 0$,*

$$\|\boldsymbol{\theta}_i^{(t)} - \boldsymbol{\theta}_i^*\|_2^2 \leq \left(1 - \frac{2\eta\sigma L}{\sigma + L}\right)^t \|\boldsymbol{\theta}_i^{(0)} - \boldsymbol{\theta}_i^*\|_2^2 \qquad \forall i \in \{1, \ldots, n\}.$$

*Proof.* This is a standard convergence result for gradient descent on strongly convex functions. For example, see Nesterov (2018). □

### C.2 Gradient descent with tied weights

**Theorem 9.** *Assume $\ell$ is $\sigma$-strongly convex and L-smooth, and consider ordinary gradient descent with tied weights*

$$\boldsymbol{\theta}_1^{(0)} = \ldots = \boldsymbol{\theta}_n^{(0)}$$
$$\boldsymbol{\theta}_i^{(t+1)} = \boldsymbol{\theta}_i^{(t)} + \eta \sum_{i=1}^n \nabla_{\boldsymbol{\theta}_i} \ell(\boldsymbol{\theta}_1, \ldots, \boldsymbol{\theta}_n) \qquad \forall i \in \{1, \ldots, n\}.$$

*There exists $\eta > 0$ such that*

$$\|\boldsymbol{\theta}_i^{(t)} - \boldsymbol{\theta}^*\|_2^2 \leq \left(1 - \frac{2\sqrt{n}\eta\sigma L}{\sigma + L}\right)^t \|\boldsymbol{\theta}_i^{(0)} - \boldsymbol{\theta}^*\|_2^2 \qquad \forall i \in \{1, \ldots, n\}.$$

*where $\boldsymbol{\theta}^* = \arg\min_{\boldsymbol{\theta}} \ell(\boldsymbol{\theta}, \ldots, \boldsymbol{\theta})$.*

*Proof.* This is again the standard convergence result, now applied to descent with step size $\sqrt{n}\eta$ on the $\sigma$-strongly convex L-smooth function $\boldsymbol{\theta} \mapsto \ell(\boldsymbol{\theta}/\sqrt{n}, \ldots, \boldsymbol{\theta}/\sqrt{n})$. Alternatively, one may think of this as projected gradient descent constrained to the subspace $\boldsymbol{\theta}_1 = \ldots = \boldsymbol{\theta}_n$ with a step size of $\sqrt{n}\eta$. Projected gradient descent inherits the same convergence properties as unconstrained gradient descent (Beck, 2017). □

### C.3 Myopic descent with untied weights

**Theorem 10.** *Assume each of the $\ell_1, \ldots, \ell_n$ are $\sigma$-strongly convex and L-smooth. Consider myopic gradient descent with untied weights*

$$\boldsymbol{\theta}_i^{(t)} = \boldsymbol{\theta}_i^{(t-1)} - \eta \nabla_{\boldsymbol{\theta}_i} \ell_i(\boldsymbol{\theta}_1^{(t-1)}, \ldots, \boldsymbol{\theta}_i^{(t-1)}) \qquad \forall i \in \{1, \ldots, n\}.$$

*There exists $\eta > 0$ such that $\boldsymbol{\theta}_i \xrightarrow{t \to \infty} \tilde{\boldsymbol{\theta}}_i$ for all $i$, where*

$$\tilde{\boldsymbol{\theta}}_1 = \arg\min_{\boldsymbol{\theta}_1} \ell_1(\boldsymbol{\theta}_1)$$

$$\tilde{\boldsymbol{\theta}}_2 = \arg\min_{\boldsymbol{\theta}_2} \ell_2(\tilde{\boldsymbol{\theta}}_1, \boldsymbol{\theta}_2)$$

$$\dots$$

$$\tilde{\boldsymbol{\theta}}_n = \arg\min_{\boldsymbol{\theta}_n} \ell_n(\tilde{\boldsymbol{\theta}}_1, \tilde{\boldsymbol{\theta}}_2, \dots, \tilde{\boldsymbol{\theta}}_{n-1}, \boldsymbol{\theta}_n).$$

*Proof.* Let $\tilde{\boldsymbol{\theta}}_1, \dots, \tilde{\boldsymbol{\theta}}_n$ be as in the theorem statement. We proceed by induction. For the base case, note that the myopic descent iterates for $\boldsymbol{\theta}_1^{(t)}$ are independent of $\boldsymbol{\theta}_j^{(t)}$ for $j > i$. Thus, the standard convergence theorem gives that $\boldsymbol{\theta}_1^{(t)} \to \tilde{\boldsymbol{\theta}}_1$ as $t \to \infty$.

Now, assume $\boldsymbol{\theta}_i^{(t)} \xrightarrow{t \to \infty} \tilde{\boldsymbol{\theta}}_i$ for all $i < k$. Thus, for any $\varepsilon > 0$, for sufficiently large $t$,

$$\|(\boldsymbol{\theta}_i^{(t)})_{i=1}^{k-1} - (\tilde{\boldsymbol{\theta}}_i)_{i=1}^{k-1}\|_2 < \varepsilon.$$

Hence, since $\ell_k$ is $L$-smooth, for any $\boldsymbol{\theta}_k$,

$$\|\nabla_{\boldsymbol{\theta}_k} \ell_k(\boldsymbol{\theta}_1^{(t)}, \dots, \boldsymbol{\theta}_{k-1}^{(t)}, \boldsymbol{\theta}_k) - \nabla_{\boldsymbol{\theta}_k} \ell_k(\tilde{\boldsymbol{\theta}}_1, \dots, \tilde{\boldsymbol{\theta}}_{k-1}, \boldsymbol{\theta}_k)\|_2 < L\varepsilon.$$

Expanding and rearranging,

$$\langle \nabla_{\boldsymbol{\theta}_k} \ell_k(\boldsymbol{\theta}_1^{(t)}, \dots, \boldsymbol{\theta}_{k-1}^{(t)}, \boldsymbol{\theta}_k), \nabla_{\boldsymbol{\theta}_k} \ell_k(\tilde{\boldsymbol{\theta}}_1, \dots, \tilde{\boldsymbol{\theta}}_{k-1}, \boldsymbol{\theta}_k) \rangle \geq \frac{1}{2} \|\nabla_{\boldsymbol{\theta}_k} \ell_k(\tilde{\boldsymbol{\theta}}_1, \dots, \tilde{\boldsymbol{\theta}}_{k-1}, \boldsymbol{\theta}_k)\|_2^2 - \frac{1}{2} L^2 \varepsilon^2$$

is bounded away from zero as long as, say,

$$\|\nabla_{\boldsymbol{\theta}_k} \ell_k(\tilde{\boldsymbol{\theta}}_1, \dots, \tilde{\boldsymbol{\theta}}_{k-1}, \boldsymbol{\theta}_k)\|_2 \geq \sigma \|\boldsymbol{\theta}_k - \tilde{\boldsymbol{\theta}}_k\|_2 > (L+1)\varepsilon,$$

using the $\sigma$-strong convexity of $\ell_k$. That is, as long as $\|\boldsymbol{\theta}_k - \tilde{\boldsymbol{\theta}}_k\|_2 > \dfrac{(L+1)\varepsilon}{\sigma}$, it is guaranteed that $-\nabla_{\boldsymbol{\theta}_k} \ell_k(\boldsymbol{\theta}_1^{(t)}, \dots, \boldsymbol{\theta}_{k-1}^{(t)}, \boldsymbol{\theta}_k)$ is a descent direction for $\ell_k(\tilde{\boldsymbol{\theta}}_1, \dots, \tilde{\boldsymbol{\theta}}_{k-1}, \boldsymbol{\theta}_k)$. This is a sufficient condition for $\boldsymbol{\theta}_k$ to converge to a $\dfrac{(L+1)\varepsilon}{\sigma}$-neighborhood of $\tilde{\boldsymbol{\theta}}_k$ given a small enough step size (Beck, 2017). Since $\varepsilon > 0$ is arbitrary, this completes the inductive step. $\square$

### C.4 Myopic descent with tied weights

**Definition 11.** *A function $f(\boldsymbol{x}, \boldsymbol{y}) \colon \mathbb{R}^a \times \mathbb{R}^a \to \mathbb{R}$ with continuous second derivatives is $\rho$-forward-biased if, for all $\boldsymbol{y} \in \mathbb{R}^a$ and $\boldsymbol{x} = \boldsymbol{y}$,*

$$\boldsymbol{H}_{\boldsymbol{y},\boldsymbol{y}} f(\boldsymbol{x}, \boldsymbol{y}) + \boldsymbol{H}_{\boldsymbol{x},\boldsymbol{y}} f(\boldsymbol{x}, \boldsymbol{y}) \succ 0$$

*and*

$$\kappa(\boldsymbol{H}_{\boldsymbol{y},\boldsymbol{y}} f(\boldsymbol{x}, \boldsymbol{y}) + \boldsymbol{H}_{\boldsymbol{x},\boldsymbol{y}} f(\boldsymbol{x}, \boldsymbol{y})) < \rho,$$

*where $\kappa$ is the condition number, and we write the Hessian of $f$ as a block matrix:*

$$\boldsymbol{H}f(\boldsymbol{x}, \boldsymbol{y}) = \begin{bmatrix} \boldsymbol{H}_{\boldsymbol{x},\boldsymbol{x}} f(\boldsymbol{x}, \boldsymbol{y}) & \boldsymbol{H}_{\boldsymbol{x},\boldsymbol{y}} f(\boldsymbol{x}, \boldsymbol{y}) \\ \boldsymbol{H}_{\boldsymbol{y},\boldsymbol{x}} f(\boldsymbol{x}, \boldsymbol{y}) & \boldsymbol{H}_{\boldsymbol{y},\boldsymbol{y}} f(\boldsymbol{x}, \boldsymbol{y}) \end{bmatrix} = \begin{bmatrix} \left( \dfrac{\partial f(\boldsymbol{x}, \boldsymbol{y})}{\partial x_i \partial x_j} \right)_{\substack{1 \leq i \leq a \\ 1 \leq j \leq a}} & \left( \dfrac{\partial f(\boldsymbol{x}, \boldsymbol{y})}{\partial x_i \partial y_j} \right)_{\substack{1 \leq i \leq a \\ 1 \leq j \leq b}} \\ \left( \dfrac{\partial f(\boldsymbol{x}, \boldsymbol{y})}{\partial y_i \partial x_j} \right)_{\substack{1 \leq i \leq b \\ 1 \leq j \leq a}} & \left( \dfrac{\partial f(\boldsymbol{x}, \boldsymbol{y})}{\partial y_i \partial y_j} \right)_{\substack{1 \leq i \leq b \\ 1 \leq j \leq b}} \end{bmatrix}.$$

**Lemma 12.** *If $\boldsymbol{A} \in \mathbb{R}^{a \times a}$ is positive definite with $\kappa(A) < \rho$, then there exists some $\eta > 0$ such that $\boldsymbol{I} - \eta \boldsymbol{A}$ is a $(1 - \rho^{-1})$-contraction. Explicitly, for any $\boldsymbol{y} \in \mathbb{R}^a$,*

$$\|(\boldsymbol{I} - \eta \boldsymbol{A})\boldsymbol{y}\|_2 \leq (1 - \rho^{-1}) \|\boldsymbol{y}\|_2.$$

*Proof.* Set $\eta = 1/\lambda_{\max}(A)$. Then $I - \eta A \succeq 0$ with

$$\lambda_{\max}(I - \eta A) = 1 - \eta\lambda_{\min}(A) = 1 - \frac{\lambda_{\min}(A)}{\lambda_{\max}(A)} < 1 - \rho^{-1}.$$

It immediately follows that $I - \eta A$ is a $(1 - \rho^{-1})$-contraction. $\square$

**Theorem 13.** *Let $f(x, y) \colon \mathbb{R}^a \times \mathbb{R}^a \to \mathbb{R}$ be $\rho$-forward biased, $\sigma$-strongly convex, and $L$-smooth with continuous second derivatives for some $\rho, \sigma, L > 0$. Then, there exists $\tilde{y} \in \mathbb{R}^a$ such that*

$$\tilde{y} = \arg\min_{y \in \mathbb{R}^a} f(\tilde{y}, y). \tag{4}$$

*Further, for some step size $\eta > 0$, the iterates of myopic gradient descent with tied weights*

$$y^{(t+1)} = y^{(t)} - \eta\nabla_y f(x, y)|_{x=y=y^{(t)}}$$

*converge to $\tilde{y} \in \mathbb{R}^a$ satisfying (4). We call such $\tilde{y}$ a myopic solution.*

We then recover the original sequential modeling setting by defining $f(\tilde{\theta}, \theta) := \sum_{i=1}^n \ell_i(\tilde{\theta}, \dots, \tilde{\theta}, \theta)$.

*Proof.* First, note that if myopic descent does converge, it must be to a point $\tilde{y}$ such that $\tilde{y} = \arg\min_y f(\tilde{y}, y)$. Indeed, at convergence we must have $\nabla_y f(\tilde{y}, y)|_{y=\tilde{y}} = 0$, so strong convexity tells us that $\tilde{y}$ is optimal. Thus, if we establish that myopic descent with small enough step size $\eta > 0$ converges to some $\tilde{y}$, we automatically get the existence of a myopic solution. For this, it suffices to show that the gradient descent step

$$g_\eta(y) = y - \eta\nabla_y f(x, y)|_{x=y}$$

is strictly contractive, so that iterates of $g_\eta$ converge to a fixed point.

Consider arbitrary $y$ and $y'$. Then, by chain rule and the mean value theorem applied to the map $y' \mapsto \nabla_y f(x, y)|_{x=y=y'}$,

$$\nabla_y f(y', y') - \nabla_y f(y, y) = (H_{x,y}f(y'', y'') + H_{y,y}f(y'', y''))(y' - y) \tag{5}$$

for some $y'' \in [y, y']$. Using the definition of $g_\eta$ and substituting in (5),

$$\begin{aligned}
\|g_\eta(y') - g_\eta(y)\|_2^2 &= \|(y' - y) - \eta(\nabla_y f(y', y') - \nabla_y f(y, y))\|_2^2 \\
&= \|(I - (H_{x,y}f(y'', y'') + H_{y,y}f(y'', y'')))(y' - y)\|_2^2 \\
&\leq (1 - \rho^{-1})(y' - y),
\end{aligned}$$

where the last line is by $\rho$-forward bias and Lemma 12. That is, $g_\eta$ is $(1 - \rho^{-1})$-contractive, completing the proof. $\square$

### C.5 Properties of sine

**Lemma 14.** *Let $x \sim \mathcal{N}(0, 1)$. Then*

$$\mathrm{Var}(\sin(bx)) = \frac{1}{2} - \frac{e^{2b^2}}{2}.$$

*Proof.*

$$\mathrm{Var}(\sin(bx)) = (2\pi)^{-1/2}\int_{-\infty}^{\infty} \sin^2(bx)e^{-x^2/2}\, dx = \frac{1}{2} - \frac{e^{2b^2}}{2}.$$

$\square$

**Lemma 15.** *Let $x \sim \mathcal{N}(0, 1)$. Then*

$$\rho(x, \sin(bx)) = \frac{2be^{3b^2/2}}{e^{2b^2} - 1} \in O(e^{-b^2/2}),$$

*where $\rho$ is the Pearson correlation coefficient.*

*Proof.* By symmetry, $\mathbb{E}[\sin(bx)] = 0$. We calculate

$$\text{Cov}(x, \sin(bx)) = (2\pi)^{-1/2} \int_{-\infty}^{\infty} x \sin(bx) e^{-x^2/2} \, dx = be^{-b^2/2}.$$

We already computed the variance in Lemma 14, so

$$\rho(x, \sin(bx)) = \frac{\text{Cov}(x, \sin(bx))}{\sqrt{\text{Var}(x)\text{Var}(\sin(bx))}} = \frac{2be^{3b^2/2}}{e^{2b^2} - 1}.$$

In our experiments we set $b = 10$, so $\rho(x, \sin(bx)) < 10^{-20}$.

## D   Details and additional experiments

### D.1   Synthetic setting: $\mathcal{D}_p$ task

We use a smaller version of the GPT-2 architecture, adapted to regression tasks. That is, the token embedding and unembedding layers are replaced with a trained linear map from the input space to the embedding space and from the embedding space to the output space, respectively. For each $p \in \{0.01, 0.1, 0.3, 1\}$ models are trained using ordinary and myopic descent on one epoch of 30M sequences of length 64 sampled from $\mathcal{D}^{64}_{p,10,10}$. See Table 3 for architecture details.

| Configuration Key | Value |
|---|---|
| num_layers | 2 |
| num_heads | 2 |
| embd_dim | 128 |
| n_inner | 512 |
| input_dim | 2 |
| output_dim | 1 |
| activation | relu |
| attn_pdrop | 0 |
| embd_pdrop | 0 |
| resid_pdrop | 0 |
| lr | 1e-3 |
| optimizer | AdamW |
| weight_decay | 0.01 |
| betas | (0.9, 0.999) |
| scheduler | cosine |
| warmup | 0.01 |
| batch_size | 512 |
| seq_length | 64 |
| loss_fn | HuberLoss |

Table 3: Transformer configuration used when training on synthetic data distribution $\mathcal{D}_p$

#### D.1.1   Probe details

Given a transformer model trained on $\mathcal{D}_p$, we sample the hidden states at each layer when the model is given as input 50 000 evaluation sequences from the same distribution $\mathcal{D}_p$. For each layer and targets $\sin(bx_{n-i})$ for varying $i > 0$, we fit a linear regression model on the hidden state of that layer to the target. The in-sample $R^2$ of each linear model is then reported in Figure 2. Figure 6 is a visualization of the linear probe's performance on the vanilla transformer.

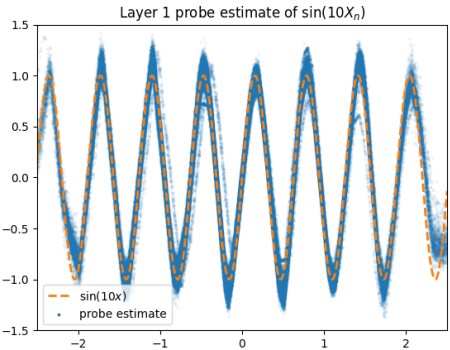

Figure 6: Estimate of $\sin(bx_n)$ by linear probe fit on layer 1 of transformer with vanilla training on $\mathcal{D}_{0.3}$. Computed over 50 000 samples from $\mathcal{D}_1$.

## D.2 Natural language setting

### D.2.1 GPT-2

For both vanilla and myopic training, we train the GPT2-small architecture from random initialization for one epoch on 4.6M sequences from the MS MARCO dataset (Nguyen et al., 2016), truncated to length 64. To estimate the local myopia bonus of the vanilla model, we train another model from random initialization with the same architecture, but with past hidden states provided by the vanilla model. Note that this "local myopic" model attains slightly *better* performance than the vanilla model; each forward pass can focus purely on next-token prediction, since past hidden states are supplied by a separate model. As a baseline, we also train a "transformer bigram" model, which is a transformer model whose key/value states are zeroed out during training and evaluation. See Table 4 for configuration details.

| Configuration Key | Value |
|---|---|
| num_layers | 12 |
| num_heads | 12 |
| embd_dim | 768 |
| n_inner | 3072 |
| vocab_size | 50257 |
| activation | gelu_new |
| attn_pdrop | 0.1 |
| embd_pdrop | 0.1 |
| resid_pdrop | 0.1 |
| lr | $6.0 \times 10^{-4}$ |
| optimizer | AdamW |
| weight_decay | 0.01 |
| betas | (0.9, 0.999) |
| scheduler | cosine |
| warmup | 0.01 |
| batch_size | 512 |
| seq_length | 64 |
| loss_fn | CrossEntropy |

Table 4: GPT-2 small configuration used when training on natural language data.

### D.2.2 Pythia

For experiments on the Pythia suite, we finetune with either vanilla or myopic descent on 10M sequences of 64 tokens each subsampled from The Pile (Gao et al., 2020). Learning rates and batch sizes for each model are presented in Table 5; they are the same between vanilla and myopic descent. All other training and architectural hyperparameters are the same as those used by Biderman et al. (2023).

| Model | Learning rate | Batch size |
|---|---|---|
| Pythia 14M | $4.0 \times 10^{-4}$ | 512 |
| Pythia 31M | $4.0 \times 10^{-4}$ | 512 |
| Pythia 70M | $1.2 \times 10^{-4}$ | 512 |
| Pythia 160M | $1.2 \times 10^{-4}$ | 512 |
| Pythia 410M | $1.2 \times 10^{-4}$ | 256 |
| Pythia 1B | $1.2 \times 10^{-4}$ | 128 |
| Pythia 1.4B | $1.2 \times 10^{-4}$ | 128 |
| Pythia 2.8B | $8.0 \times 10^{-5}$ | 64 |

Table 5: Pythia suite hyperparameters for finetuning. Batch size is measured in sequences of 64 tokens each.

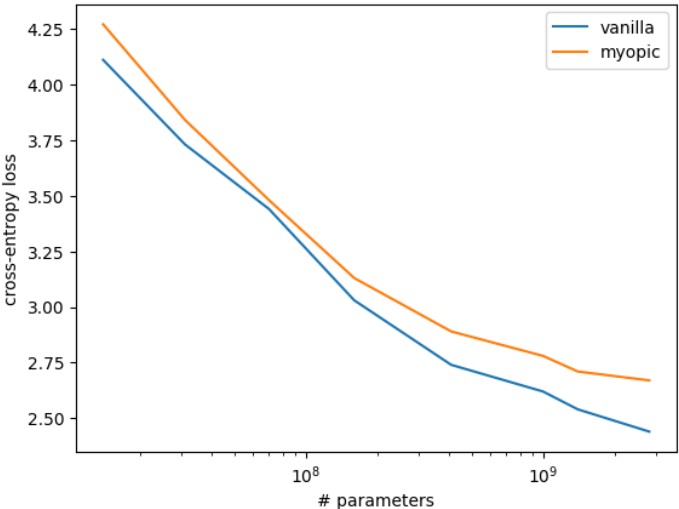

Figure 7: Cross-entropy loss of Pythia models fine-tuned on The Pile dataset using vanilla and myopic gradient descent. Starting from the 70M model, we see that the gap increases with parameter count.

### D.3 Multiplication

In addition to the natural language experiments, we also measure the myopia gap on the task of natural number multiplication. We use the same GPT2-small architecture as in the natural language experiments starting from random initialization; see Table 4. See Figure 8 for an example input sequence.

$$3\ 7\ 0\ 0\ *\ 5\ 4\ 0\ 0\ =\ 5\ 8\ 2\ 3\ 0\ 0\ 0\ 0$$

Figure 8: Example multiplication input sequence.

|   | 1 | 2 | 3 | 4 | 5 | 6 | 7 | 8 |
|---|---|---|---|---|---|---|---|---|
| 1 | 1.00 | 1.00 | 1.00 | 1.00 | 1.00 | 1.00 | 1.00 | 1.00 |
| 2 | 1.00 | 1.00 | 1.00 | 1.00 | 1.00 | 1.00 | 1.00 | 1.00 |
| 3 | 1.00 | 1.00 | 1.00 | 1.00 | 1.00 | 1.00 | 1.00 | 1.00 |
| 4 | 1.00 | 1.00 | 1.00 | 1.00 | 1.00 | 1.00 | 1.00 | 0.99 |
| 5 | 1.00 | 1.00 | 1.00 | 1.00 | 0.98 | 0.92 | 0.21 | 0.10 |
| 6 | 1.00 | 1.00 | 0.99 | 0.99 | 0.59 | 0.16 | 0.02 | 0.02 |
| 7 | 1.00 | 1.00 | 1.00 | 0.96 | 0.16 | 0.03 | 0.00 | 0.00 |
| 8 | 1.00 | 1.00 | 0.99 | 0.51 | 0.06 | 0.01 | 0.00 | 0.00 |

Vanilla multiplication accuracy

|   | 1 | 2 | 3 | 4 | 5 | 6 | 7 | 8 |
|---|---|---|---|---|---|---|---|---|
| 1 | 1.00 | 1.00 | 1.00 | 1.00 | 1.00 | 1.00 | 1.00 | 1.00 |
| 2 | 1.00 | 1.00 | 1.00 | 1.00 | 1.00 | 1.00 | 1.00 | 1.00 |
| 3 | 1.00 | 1.00 | 1.00 | 1.00 | 1.00 | 1.00 | 1.00 | 1.00 |
| 4 | 1.00 | 1.00 | 1.00 | 1.00 | 0.70 | 0.50 | 0.34 | 0.26 |
| 5 | 1.00 | 1.00 | 1.00 | 0.99 | 0.45 | 0.16 | 0.07 | 0.03 |
| 6 | 1.00 | 1.00 | 1.00 | 0.96 | 0.27 | 0.05 | 0.01 | 0.00 |
| 7 | 1.00 | 1.00 | 1.00 | 0.60 | 0.11 | 0.02 | 0.00 | 0.00 |
| 8 | 1.00 | 1.00 | 1.00 | 0.41 | 0.05 | 0.00 | 0.00 | 0.00 |

Myopic multiplication accuracy

|   | 1 | 2 | 3 | 4 | 5 | 6 | 7 | 8 |
|---|---|---|---|---|---|---|---|---|
| 1 | 0.00 | 0.00 | 0.00 | 0.00 | 0.00 | 0.00 | 0.00 | 0.00 |
| 2 | 0.00 | 0.00 | 0.00 | 0.00 | 0.00 | 0.00 | 0.00 | 0.00 |
| 3 | 0.00 | 0.00 | 0.00 | 0.00 | 0.00 | 0.00 | 0.00 | -0.00 |
| 4 | 0.00 | 0.00 | 0.00 | 0.00 | 0.30 | 0.50 | 0.65 | 0.73 |
| 5 | 0.00 | 0.00 | -0.00 | 0.01 | 0.53 | 0.77 | 0.13 | 0.07 |
| 6 | 0.00 | 0.00 | -0.01 | 0.03 | 0.32 | 0.11 | 0.01 | 0.02 |
| 7 | 0.00 | 0.00 | -0.00 | 0.36 | 0.05 | 0.01 | -0.00 | 0.00 |
| 8 | 0.00 | 0.00 | -0.01 | 0.10 | 0.01 | 0.01 | 0.00 | 0.00 |

Vanilla-myopic accuracy gap

Figure 9: Accuracy of vanilla and myopic transformers trained on multiplication of up to 8-digit inputs. Row and columns correspond to the number of digits in the first and second multiplicands, respectively.

We use several formatting tricks found by Shen et al. (2023) to improve performance on the multiplication task:

- Characters are delimited by spaces, such that each digit is tokenized into a separate token.

- All numbers are written in the reverse of the standard order, i.e. such that the least significant digits come first.

- All inputs are zero-padded to the same length.

Note for each multiplicand we first sample the number of digits $d \sim \text{Unif}(n)$ uniformly in some range $n$, then uniformly sample a natural number $x \sim \text{Unif}(10^d - 1)$ with no more than $d$ digits. This distribution allocates increased weight to smaller numbers, and was found to result in superior performance.

We train both vanilla and myopic transformers on one epoch of 10M examples with no more than 8 digits, then measure 0/1 accuracy (that is, the model is provided with the an input sample up to the '=' token, and scored 1 if it completes the rest of the sequence exactly correctly and 0 otherwise) on 1024 independent random validation examples for each of the $8 \times 8$ possible pairs of digit counts for the two multiplicands. See Figure 9.

### D.3.1 Filler tokens

We further hypothesize that, as in Pfau et al. (2024), the vanilla transformer may learn to perform computation even on forward passes corresponding to filler tokens, thus attaining better performance when trained on examples zero-padded to longer lengths. We expect that myopic transformers, on the other hand, are not incentivized to do this, since this extra computation holds no relevance towards the immediate task of predicting the filler zero token. To test this hypothesis, we train vanilla and myopic transformers on each of two different multiplication datasets:

1. Both multiplicands have at most 5 digits, and are zero-padded to exactly 5 digits.

2. Both multiplicands have at most 5 digits, and are zero-padded to exactly 10 digits.

Again, all training runs consist of one epoch of 10M examples.

See Figure 10 for results. Note that the vanilla transformer indeed performs better when trained and evaluated on input sequences zero-padded to a longer length. However, the myopic transformer performs substantially worse with increased zero-padding. Our explanation is that, not only does the myopic transformer not learn to perform intermediate tokens during zero-token forward passes, the increased input length makes it more difficult for the attention mechanism to correctly attend to the relevant tokens.

### D.4 Gradient angles

Using the publicly available training checkpoints for Pythia-410M (Biderman et al., 2023), we measure the sizes of both the myopic component and the future component of the gradient of the loss w/r/t the parameters over the course of training. (Note that the future component is the difference between the total vanilla gradient and the myopic gradient.) We also measure the cosine similarity between the myopic and future components. See Figure 11. One observation is that the norm of the myopic gradient is consistently larger than that of the future gradient—thus, training is dominated by the effect of each forward pass's parameters on the immediate next-token prediction.

|   | 1 | 2 | 3 | 4 | 5 |
|---|---|---|---|---|---|
| 1 | 1.00 | 1.00 | 1.00 | 1.00 | 1.00 |
| 2 | 1.00 | 1.00 | 1.00 | 1.00 | 1.00 |
| 3 | 1.00 | 1.00 | 1.00 | 1.00 | 1.00 |
| 4 | 1.00 | 1.00 | 1.00 | 1.00 | 1.00 |
| 5 | 1.00 | 1.00 | 1.00 | 1.00 | 0.40 |

Vanilla accuracy, padded to length 5

|   | 1 | 2 | 3 | 4 | 5 |
|---|---|---|---|---|---|
| 1 | 1.00 | 1.00 | 1.00 | 1.00 | 1.00 |
| 2 | 1.00 | 1.00 | 1.00 | 1.00 | 1.00 |
| 3 | 1.00 | 1.00 | 1.00 | 1.00 | 1.00 |
| 4 | 1.00 | 1.00 | 1.00 | 1.00 | 1.00 |
| 5 | 1.00 | 1.00 | 1.00 | 1.00 | 0.99 |

Vanilla accuracy, padded to length 10

|   | 1 | 2 | 3 | 4 | 5 |
|---|---|---|---|---|---|
| 1 | 1.00 | 1.00 | 1.00 | 1.00 | 1.00 |
| 2 | 1.00 | 1.00 | 1.00 | 1.00 | 1.00 |
| 3 | 1.00 | 1.00 | 1.00 | 1.00 | 1.00 |
| 4 | 1.00 | 1.00 | 1.00 | 0.98 | 0.79 |
| 5 | 1.00 | 1.00 | 1.00 | 0.82 | 0.39 |

Myopic accuracy, padded to length 5

|   | 1 | 2 | 3 | 4 | 5 |
|---|---|---|---|---|---|
| 1 | 1.00 | 1.00 | 1.00 | 1.00 | 1.00 |
| 2 | 1.00 | 1.00 | 1.00 | 1.00 | 1.00 |
| 3 | 1.00 | 1.00 | 1.00 | 1.00 | 0.99 |
| 4 | 1.00 | 1.00 | 1.00 | 0.43 | 0.24 |
| 5 | 1.00 | 1.00 | 0.97 | 0.22 | 0.05 |

Myopic accuracy, padded to length 10

Figure 10: Multiplication accuracy of GPT-2 with either vanilla or myopic training and with input multiplicands zero-padded to either length 5 or 10. Row and columns correspond to the number of digits in the first and second multiplicands, respectively. Observe that padding improves performance of the vanilla model, but decreases performance of the myopic model.

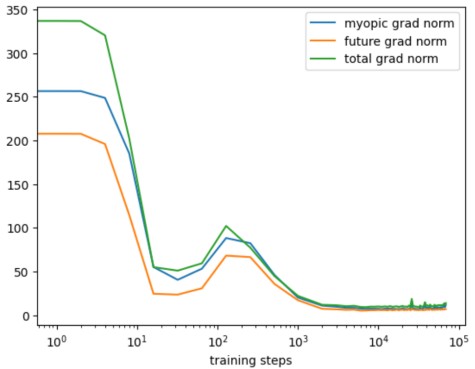 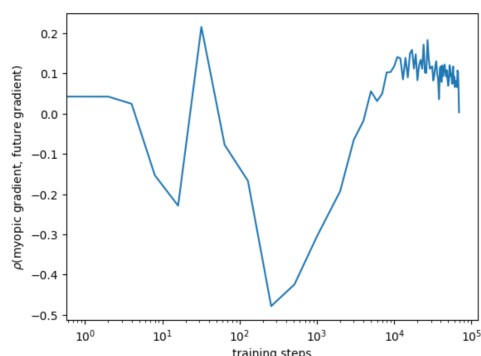

Norms of myopic, future, and total gradients    Cosine similarity between myopic and future gradient

Figure 11: Myopic and future gradients of Pythia-410M during training.

