# OpenReview forum: "Do Language Models Plan Ahead for Future Tokens?"
_colmweb.org/COLM/2024/Conference — COLM_

### Official Review · Reviewer_ZSJf · 2024-05-01

**Rating:** 7
**Confidence:** 3
**Ethics Flag:** 1

**Summary:**

This paper studies whether Transformer language models encode information in middle layers that is useful for future token predictions, or are they mostly "greedy" in a sense that they learn features relevant for the current token prediction and these feature end up being useful for future predictions. The authors formalize this research question and call the two hypotheses "pre-caching" and "breadcrumbs" respectively. They go ahead and design experiments for this question by avoiding the gradient updates to previous timesteps and measuring the gap in perfomance between optimal models trained with past awareness and without. On a synthetic dataset that requires continuously memorizing the past for predicting the next token, the authors find evidence for pre-caching. On small scale LM training, it appears that the breadcrumbs case is more likely.

**Questions To Authors:**

Other than studying the behavior of models, do you think there are any practical conclusions? for example, if the breadcrumbs hypothesis is true, maybe some more efficient pretraining schema is possible?

**Reasons To Accept:**

An interesting study on the dynamics of language modeling training with nice formulations and experimental setup to create feasible but interesting explorations.

**Reasons To Reject:**

Overall I think the study is interesting and worth sharing with the community. Yet, I would note that the LM experiments are on only 124M GPT-2 models 4.6M texts of length 64 from MS MARCO. This is much smaller in model size, context length and token scale compared to today's LLMs. So the conclusion of natural language learning following the breadcrumbs hypothesis might only hold for this limited studied setting and not hold for realistic settings. I'd be curious to know this, but given the cost of such an experiment I understand that it might be infeasible to include.

---

> ### Author Rebuttal · Authors · 2024-05-30
>
> Thank you for your review.
>
> > I would note that the LM experiments are on only 124M GPT-2 models 4.6M texts of length 64 from MS MARCO.
>
> The final revision will contain further myopia gap experiments on models from the Pythia suite ranging up to 2.8B parameters in size, evaluated on both cross-entropy and various NLP benchmarks.
>
> > Other than studying the behavior of models, do you think there are any practical conclusions?
>
> We hypothesize that myopic transformers may have advantages in terms of safety and/or interpretability — it may be easier to understand what a model is doing if we know that everything that it is computing on each forward pass is directly towards the goal of predicting the immediate next token.
>
> One manifestation of this is our finding (which will be included in the final version) that pre-caching is necessary to make use of computation on the forward passes of filler tokens, in the sense of Pfau et al. "Let's think dot by dot". We train and evaluate vanilla and myopic transformers on an integer multiplication task. Our experiments show that vanilla transformers perform better on this task given more zero-padding tokens, while myopic transformers actually do worse. We interpret this as showing that pre-caching is necessary for making effective use of these filler tokens — without pre-caching, these filler tokens are mere noise, and actually degrade performance.
>
> Another possibility, which we have not yet explored, is that of automatically swapping in a myopic model on forward passes where we detect it is beneficial to sacrifice future performance in favor of immediate next-token accuracy (for example, on especially important tokens, or near the end of a text).
>
> Also, if indeed the myopia gap remains acceptably small for production-sized models, there may be some efficiency gains to be had be training myopically, and thus not computing the off-diagonal gradient terms. Again, we have not yet investigated this possibility, and leave it to future work.

---

> > ### Comment · Reviewer_ZSJf · 2024-06-07
> >
> > thank you for your response

---

### Official Review · Reviewer_4NSr · 2024-05-07

**Rating:** 6
**Confidence:** 4
**Ethics Flag:** 1

**Summary:**

This paper investigates whether the hidden states of large autoregressive language models encode information solely useful to predict future tokens (pre-caching hypothesis), or whether any such information is a side effect of being able to predict the following word accurately (breadcrumb hypothesis).

It proposes a quantity called the myopia gap: the difference in loss between a globally optimal LM and one where the parameters at each time step are greedily optimized for the loss at that position alone. It proposes to estimate the myopia gap by comparing the loss of a typically trained model, and one where the token-level loss is backpropagated only with respect to the weights at the current position. While weights at each time step are tied, the paper engages in a thought experiment, first supposing parameters to be untied across time steps, and develops "myopic gradient descent" to optimize LM parameters only for predicting the subsequent position.

The paper constructs a synthetic data set, and demonstrates that the myopia gap of a 2-layer transformer trained on this data is high (by construction). By applying a linear probe, it also demonstrates that the representation of the first hidden layer is strongly correlated with a value consistent with the pre-caching hypothesis, which is not true if the model is trained myopically. Training an $\approx$100M parameter LM on MS MARCO shows that the myopia gap exists for larger LMs, although the difference is slight, and is a function of the token position as well.

**Questions To Authors:**

- One may consider discussing work on blockwise parallel [1]/non-autoregressive decoding, which explicitly train models to predict a block of subsequent tokens in parallel.
- Why use PG-19 for evaluation in Figure 5 when MS MARCO was used as the training set? Wouldn't this be conflating domain shift with general LM quality?
- What are the practical implications of whether the model is employing pre-caching vs. breadcrumbs, conditioned on similar model performance? Is one a priori more desirable?
- Why does the myopically-trained model incur lower loss on the first few token positions than the standard LM (Figure 5)?
- Appendix C: tilde is used inconsistently (e.g., when defining Q, K, V, and also over X).
- Appendix G: This appears to be orthogonal to the experiments in the body text and is never referred to from the body. It is unclear how this relates to the myopia gap, or if this was a separate effort at understanding the tendency of LLMS to pre-cache.

[1] Stern, Mitchell, Noam Shazeer, and Jakob Uszkoreit. "Blockwise parallel decoding for deep autoregressive models." Advances in Neural Information Processing Systems 31 (2018).

**Reasons To Accept:**

The concept of "myopia gap" may be of interest to LLM researchers and this paper makes a valiant effort in justifying its use as a tool for understanding what information LLMs acquire during training. The paper shows that myopic training of LMs indeed results in a high myopia gap in a synthetic data scenario. While the experiments on larger LMs are light, the concept of myopia gap could be a useful tool for understanding how more realistically-sized/trained LLMs operate.

**Reasons To Reject:**

It is unclear from the paper what constitutes a significant amount of myopia gap. How does one test the hypothesis that a model is explicitly using pre-caching? The paper does not answer this explicitly, just alludes to the fact that myopically trained model incurs a penalty of $\approx0.12$ in terms of cross-entropy loss. In addition, the actual model ($\approx$100M parameters) and training data ($\approx$250M tokens) sizes are modest. It is unclear whether the natural language findings regarding myopia gap would generalize to more realistic LLMs, trained to convergence on larger corpora.

---

> ### Author Rebuttal · Authors · 2024-05-30
>
> > the actual model and training data are modest.
>
> The final revision will contain further myopia gap experiments on models ranging up to 2.8B parameters in size.
>
> > It is unclear what constitutes a significant amount of myopia gap
>
> The myopia gap of $\sim 0.12$ is best understood in the context of relative comparisons. Eg this gap is much smaller than that between the standard transformer and the “transformer bigram” referenced in the text. The aforementioned new experiments will also give more points of comparison between models of different scale. We will also include results on NLP benchmarks, which may more meaningfully translate to performance on real-world tasks
>
> > One may consider discussing work on blockwise parallel /non-autoregressive decoding
>
> We will add the suggested citation. Note that we also cite Cai et al. "Medusa". We will expand our discussion of these related works in the final revision
>
> > Why use PG-19 for evaluation in Figure 5
>
> The PG-19 dataset includes longer texts, and thus allows us to evaluate using a rolling window over a single text. This reduces noise in the per-position loss plot, as every token is used in every position. The domain shift between MS MARCO and PG-19 does not affect our findings---what we care about is the relative comparison between the vanilla and myopic models, both evaluated on PG-19
>
> > What are the practical implications of whether the model is employing pre-caching vs. breadcrumbs?
>
> One possibility is that of automatically swapping in a myopic model on forward passes where we detect it is beneficial to sacrifice future performance in favor of immediate next-token accuracy (for example, on especially important tokens, or near the end of a text).
> Also, if indeed the myopia gap remains acceptably small for production-sized models, there may be some efficiency gains to be had be training myopically, and thus not computing the off-diagonal gradient terms.
> We leave these directions to future work
>
> > Why does the myopically-trained model incur lower loss on the first few token positions than the standard LM (Figure 5)?
>
> Intuitively, the myopic model can dedicate all of its compute on the immediate next token. This advantage outweighs the cost of not pre-caching for the first few tokens
>
> > Appendix G: This appears to be orthogonal to the experiments in the body text
>
> We agree that Appendix G would make more sense as part of a separate work. It will be removed in the final revision
>
> > Appendix C: tilde
>
> Will fix

---

### Official Review · Reviewer_URPJ · 2024-05-17

**Rating:** 7
**Confidence:** 3
**Ethics Flag:** 1

**Summary:**

The paper investigates the idea of a transformer-based LM taking decisions by "planning ahead" (pre-caching) or merely making use of prior states that were deemed optimal for the prior decisions (breadcrumbs). They do this by implementing a reduced "myopic" transformer which does not update prior states during updates with later decisions during training.

The authors perform a synthetic experiment which shows that a myopically restricted model is unable to pre-cache prior results whereas a standard transformer does for this specifically designed synthetic task.
The authors then train both kinds of models as well as a model which strictly limits attention to bigramming on natural language data and find that the myopic model performs much closer to the standard than the bigram model. The authors take this as an indication that a transformer is largely breadcrumbing and that the performance increase from using a model that is able to pre-cache is only moderate, which leads to the conclusion that little pre-caching is going on.

**Questions To Authors:**

See above. Some remarks:
- Figure 4 is not referenced from the text. (It an also probably be omitted.) I'd put Table 1 earlier, into Section 4.
- When you refer to Table 4, it would help to remark that it can be found in Appendix F.
- There's an Appendix G which is not referenced from the main text. What does it do?
- I share your belief that humans "pre-cache" in some way. However, is there reason to believe that boosting this behaviour (beyond what happens in your standard transformer) would lead to better language modelling results? If so, in what ways? Would it be possible to design corresponding follow-up work? Can you measure this in ways that are different from cross-entropy? (E.g., would it help an LM to pre-compute when it sees the second addend rather than think about what type of second object (apple) it is?

**Reasons To Accept:**

valid analysis of what a transformer does internally.
definition and thorough discussion of myopic training that could be useful in future analyses

**Reasons To Reject:**

The implications of this research, in particular whether a model might be pre-caching or breadcrumbing remain open.
The paper would be much more approachable and engaging if it could elaborate on what's hinted at in Figure 1 (which is not referenced from the text!), i.e., that humans start pondering about computations (and the like) in a more structured form. Is there also evidence that this is helpful? Maybe could one come up with ways of training "focally" instead of "myopically", i.e. lay caches in certain ways that then yield better results?

The paper is nice theoretically but I would like to see a closer discussion of NLP topics. For example, when you discuss future token meta-prediction, it would help to highlight that of course next tokens can be predicted from state as that's what's the whole point of language modelling. I presume that the cited works (nostalgebraist is an incomplete citation!) take precautions (synthetic data or in the evaluation?) that future-token prediction doesn't simply stem from the fact that a next-to-next token will clearly depend on the next token and can hence be predicted from the next token's hidden state with some accuracy.

---

> ### Author Rebuttal · Authors · 2024-05-30
>
> Thank you for your review.
>
> > Maybe could one come up with ways of training "focally" instead of "myopically"
>
> It is in principle possible to train with up-weighted off-diagonal gradient terms, thus resulting in a “hyperopic” model. We leave this to future work.
>
> > I would like to see a closer discussion of NLP topics.
>
> > Can you measure this in ways that are different from cross-entropy?
>
> The final revision will contain more experiments on a range of models up to 2.8B params in size, evaluated on standard NLP benchmarks
>
> > it would help to highlight that of course next tokens can be predicted from state
>
> We cite Pal et al. "Future Lens", which investigates this claim empirically.
>
> > I presume that the cited works take precautions that future-token prediction doesn't simply stem from the fact that a next-to-next token will clearly depend on the next token
>
> The cited Pal et al. paper accounts for this by measuring the effect of causal interventions on the hidden state.
>
> > Figure 4 is not referenced from the text.
>
> We will fix this. Fig 4 demonstrates the linear probe’s ability to detect $\sin(bx_n)$ in the hidden state.
>
> > When you refer to Table 4, it would help to remark that it can be found in Appendix F.
>
> We will do this.
>
> > There's an Appendix G which is not referenced from the main text.
>
> We agree that Appendix G would make more sense as part of a separate work. It will be removed in the final revision.
>
> > However, is there reason to believe that boosting this behaviour (beyond what happens in your standard transformer) would lead to better language modelling results?
>
> We do not claim that boosting pre-caching would benefit performance. However, this is an interesting question to investigate in future work.
>
> > The implications of this research, in particular whether a model might be pre-caching or breadcrumbing remain open.
>
> We agree that much future work can be done investigating the practical implications of pre-caching vs breadcrumbing. One possibility is that of automatically swapping in a myopic model on forward passes where we detect it is beneficial to sacrifice future performance in favor of immediate next-token accuracy (for example, on especially important tokens, or near the end of a text).
>
> Also, if indeed the myopia gap remains acceptably small for production-sized models, there may be some efficiency gains to be had by training myopically, and thus not computing the off-diagonal gradient terms.
> We leave these directions to future work.

---

### Author Response · Authors · 2024-05-31
**To all reviewers: details of new experiments**

Thanks to all reviewers. Here are some additional details on the new results we intend to include in the final revision of the paper. Note that all experiments have already been run, but we only include a subset of them in this comment due to space considerations.

**Scaling of myopia gap on Pythia family**

We investigate the myopia gap on a range of model sizes from the Pythia family, ranging from 70M to 2.8B parameters. In particular, we are interested in how the myopia gap scales with model size — multiple reviewers suggested that the insignificant gap observed in GPT-2 may be explained by it being a relatively small model. Our experiments reveal a trend towards myopia gap increasing with scale; this suggests that pre-caching is indeed more explanative of the behavior of larger models. Further, we measure the gap not only on validation cross-entropy loss but also a suite of standard NLP benchmarks (LAMBADA, PIQA, WinoGrande, SciQ, ARC), which may be a more realistic measure of the role of pre-caching in real-world language tasks. Comparisons of the myopia gap between these benchmarks may offer insight into what kinds of tasks benefit more from pre-caching.

As a sample, here are the results on the LAMBADA benchmark [1]:

size | 14M | 31M | 70M | 160M | 410M | 1B | 1.4B | 2.8B
---  | --- | --- | --- | ---  | ---  | -- | ---  | ---
vanilla accuracy | 0.06 | 0.10 | 0.19 | 0.28 | 0.43 | 0.47 | 0.54 | 0.59
myopic  accuracy | 0.02 | 0.07 | 0.13 | 0.21 | 0.29 | 0.35 | 0.38 | 0.40
myopia gap       | 0.04 | 0.03 | 0.06 | 0.07 | 0.14 | 0.12 | 0.16 | 0.19

Note that, unlike in the case of GPT-2, we are unable to train the Pythia models from random initialization (due to limitations in compute). Thus, our methodology is to start from the provided pre-trained checkpoints, and to fine-tune using either vanilla or myopic gradient descent on a further 640M tokens subsampled from the Pile dataset.

**Multiplication and filler tokens**

We also train and evaluate the GPT-2 architecture, using either vanilla or myopic gradient descent, on the task of integer multiplication. We again observe a subtle but nonzero myopia gap in this setting.

Furthermore, we investigate the role of pre-caching in computations on the forward pass of filler tokens, in the sense of Pfau et al [2]. We train vanilla and myopic transformers on multiplication of up to five-digit numbers, but with each multiplicand zero-padded either to length 5 or length 10. Our experiments show that vanilla transformers perform better on this task given more zero-padding tokens, while myopic transformers actually do worse. We interpret this as showing that pre-caching is necessary for making effective use of these filler tokens — without pre-caching, these filler tokens are mere noise, and actually degrade performance. The below table contains the accuracies of both models on multiplication of two five-digit multiplicands, each either zero-padded to length 5 or length 10.

model | pad5 acc. | pad10 acc.
-- | -- | --
vanilla | 0.40 | 0.99
myopic  | 0.39  | 0.05

[1] Paperno et al. "The LAMBADA dataset: Word prediction requiring a broad discourse context."

[2] Jacob Pfau, William Merrill, Samuel R. Bowman. “Let's Think Dot by Dot: Hidden Computation in Transformer Language Models.”

---

### Decision · Program_Chairs · 2024-07-10

**Decision:**

Accept

**Comment:**

This paper analyses if/how ("small", 124M) LLMs plan ahead, i.e. their hidden state contains information useful for only the next token (breadcrumbing), or for future time steps (pre-caching).
Experiments with modified/myopically restricted model unable to pre-cache and a standard transformer on a synthetic task are used to shed light on this, but do leave many open questions.

All reviewers (URPJ, 4NSr, ZSJf) were positive in general, and mentioned the "valid analysis" (URPJ), say that the myopia gap is of interest (4NSr), and that it is "interesting", and that the experimental setup is good (ZSJf). The critiques are the that there could be more NLP discussion (URPJ), that the scale is too small for broad conclusions (4NSr, ZSJf), which was countered by the authors providing extra results in the rebuttal phase as well.

All in all the paper makes for a good contribution despite the (initial) small scale, so in my view the pros outweigh the cons, especially since some weaknesses could be mitigated for camera ready.

[comments from PCs] Please refine and incorporate the results mentioned by the AC.